# Evaluation of Effectiveness and Safety of Microcin C7 in Weaned Piglets

**DOI:** 10.3390/ani12233267

**Published:** 2022-11-24

**Authors:** Lijun Shang, Junyan Zhou, Jiayu Tu, Xiangfang Zeng, Shiyan Qiao

**Affiliations:** 1State Key Laboratory of Animal Nutrition, Ministry of Agriculture Feed Industry Centre, China Agricultural University, Beijing 100193, China; 2Beijing Bio-Feed Additives Key Laboratory, Beijing 100193, China

**Keywords:** antimicrobial peptide, antibiotic substitute, piglet, growth performance, diarrhea, microbiota

## Abstract

**Simple Summary:**

The bactericidal mechanisms and immunomodulatory effects of Microcin C7 (C7) have been confirmed by many studies. However, most studies were performed in vitro and need to be substantiated with in vivo data. We showed that C7 supplementation can improve growth performance, reduce diarrhea rate, and that the diarrhea-alleviating effect of C7 may be related to its selective regulation of specific microbial taxa. C7 improved the intestinal morphology of the duodenum and ileum and enhanced diet digestibility. The recommended C7 dosage is 250–500 mg/kg. Supplementation with 5000 mg/kg had no adverse effect on all indices; thus, this is a safe threshold for C7 usage in clinical practice. Overall, these data indicate that C7 is safe and effective for use as a potential alternative to antibiotic growth promoters in weaned piglets.

**Abstract:**

The effects and safety of dietary supplementation with Microcin C7 (C7) were evaluated in 216 weaned piglets. The pigs were given a control corn–soybean meal basal diet or C7 diet (control diet supplemented with 250, 500, 750, 1000, or 5000 mg C7/kg diets). Compared with the control group, the 500 mg/kg C7 supplementation group had better intestinal morphological indicators (*p* < 0.05), which may help maintain intestinal epithelial function and increase the growth performance (*p* < 0.05) and apparent total tract digestibility (*p* < 0.05). The diarrhea indexes of the 250, 500, and 750 mg/kg groups were significantly lower than that of the control group at 0–28 d (*p* < 0.05), and the 500 mg/kg group had the lowest diarrhea indexes (linear and quadratic, *p* < 0.05). A comprehensive analysis showed that microbial structure was significantly correlated with the degree of diarrhea, and the diarrhea-alleviating effect of C7 may be related to its selective regulation of specific microbial taxa. The 250 and 500 mg/kg C7 supplementation also significantly improved several immune indices of piglets (*p* < 0.05). Compared with the control diet, 5000 mg/kg C7 supplementation had no significant adverse effect on all parameters. Overall, the 250–500 mg/kg dose had the best effect, and the highest dose (5000 mg/kg) posed no toxicity risk. Therefore, C7 appears safe for use as an alternative to antibiotic growth promoters in weaned piglets.

## 1. Introduction

Weaning exposes piglets to various stresses such as feed changes and separation from sows, and its negative effects are mainly related to gastrointestinal disorders caused by physiological, immunological, and microbial changes in the gastrointestinal tract [1,2]. Weaning is therefore seen as a stressful event that results in decreased feed intake and growth performance and increased disease and mortality, and thus causes substantial economic losses in the pig industry [3,4]. As a result, in-feed antibiotics have long been used to maintain piglet health and improve growth performance [4,5]. However, the continued use and abuse of antibiotics has led to the emergence of drug resistance [6] together with an increased risk of antibiotic residues accumulating in animal products [7], which has become a major human health issue. This has led to a need for the development of effective alternatives to antibiotics that are highly effective, less toxic, and can be used to boost the host defense system of weaned piglets.

In recent years, considerable efforts have been made in the development of novel alternatives to antibiotic feed additives. The desired alternatives should improve the intestinal health of animals without promoting bacterial resistance, thereby reducing economic losses in pig production and producing products that meet environmental and health needs. As antibiotic replacements, synthetic natural antimicrobial peptides (AMPs) are considered ideal candidates [8,9]. AMPs are defined as “evolutionarily conserved short cationic molecules that present in all living organisms and support the host’s defense against microbial infections” [10]; they have immunoregulatory activity [11,12,13], natural antimicrobial properties, and difficultly developing bacterial resistance [8,9,14]. AMPs have proven successful in preventing diarrhea and improving growth performance by immunomodulatory or bactericidal means; nevertheless, the benefits tend to be peptide-specific [15,16,17,18,19].

Microcin C7 (C7) is a Trojan horse AMP that mimics the aminoacyl–adenylate intermediate and targets and inhibits aspartyl tRNA synthetase. Previous studies have shown that C7 at nanomolar concentrations exhibits antimicrobial activity against gram-negative strains [20]. Therefore, the extremely strong bactericidal activity combined with its safe and stable properties makes it a practical alternative to antibiotics that can be applied in the animal husbandry industry. However, little is known about its efficacy at present. Previous studies have focused on its bactericidal mechanisms, immunomodulatory effects, and production with most of the work being conducted in vitro [21,22,23,24,25,26]. The use of C7 as a feed additive is still in its infancy and very few animal studies have been documented. However, as with any potential antibiotic alternatives, in vitro observations need to be confirmed by in vivo data. Therefore, this paper looks at the efficacy and safety of C7 as feed additives in weaned piglets. Its effectiveness was reflected by growth performance, diarrhea rate, apparent total tract digestibility (ATTD) of nutrients, serum immunoglobulin levels, and intestinal morphology, and the optimal supplementary dosage was determined according to the above indicators. Fecal samples were collected to examine the effect of C7 on the microbial community of weaned piglets and to analyze whether the altered microbiota composition correlated with its beneficial effects. A high dose group was set up to explore the safety of C7 in weaned piglets. Key parameters, including growth performance; organ weights; small intestine, liver, and kidney pathology; and routine blood and blood biochemistry were investigated to evaluate the safety of C7 in vivo.

## 2. Materials and Methods

All experimental procedures and animal care were approved by the China Agricultural University Animal Care and Use Committee (Beijing, China).

### 2.1. Animals and Experimental Design

A total of 216 Duroc × Yorkshire × Landrace weaned piglets (7.90 ± 0.10 kg) were used in a 28 d performance trial. The experiments were conducted at the FengNing Swine Research Unit of China Agricultural University (Chengdejiuyun Agricultural and Livestock Co., Ltd., Chengde, China). The pigs were randomly allotted to one of six treatments with six replicates per treatment and six pigs per pen. A corn–soybean meal basal diet was formulated based on NRC 2012 (Table 1). Five additional diets were formulated by adding 250, 500, 750, 1000, or 5000 mg/kg C7 into the control diet (the measured proportion of the active ingredient Microcin C7 was 1.19% and the rest was corncob powder and medical stone carrier at a ratio of 2:1, provided by AGELESS (CHONGQING) BIO-TECH Co., Ltd., Chongqing, China).

### 2.2. Performance and Incidence of Diarrhea

Body weight (BW) and feed intake were recorded at d 0, d 14, and d 28. On the basis of these measurements, the average daily gain (ADG), average daily feed intake (ADFI), and feed-to-gain ratio (F:G) were calculated.

Twice a day, a single trained person evaluated the fecal scores based on stool classification in the pen. The scoring system is displayed in Table 2, and the formulae used were as follows:Diarrhea index = sum of diarrhea score/(number of tested piglets × trial days)
Diarrhea frequency = number of diarrhea piglets/(number of tested piglets × trial days)

### 2.3. Apparent Total Tract Digestibility Measurements

From d 25 to d 28, about 100 g of feces were collected and stored at −20 °C before oven drying. The samples were dried at 65 °C for 72 h, and then were ground and passed through a 1 mm screen (40 mesh) before analysis.

Feed and fecal samples were analyzed in terms of crude protein (CP), gross energy (GE), ether extract (EE), organic matter (OM), Ca, P, and dry matter (DM) with an automatic isoperibol oxygen bomb calorimeter (Parr 1281, Automatic Energy Analyzer; Moline, IL, USA). Digestibility was calculated by measuring chromium content in the diets and feces (Atomic Absorption Spectrophotometer; Z-5000, Hitachi, Tokyo, Japan).

### 2.4. Blood Sampling and Analysis

Blood samples were collected by venipuncture of the anterior vena cava from six randomly selected pigs in each treatment group on d 14 and d 28. Samples were maintained at room temperature for 2–3 h prior to centrifugation (3000× *g* for 10 min), and the serum was obtained and stored at −80 °C. Whole-blood samples from the control, 500 mg/kg, and 5000 mg/kg groups were collected in EDTA tubes and used for hematology analysis within 6 h after sampling.

The serum immune indexes for IgA, IgG, IgM, IL-1β, TNF-α, IFN-γ, IL-6, IL-10, CD3+, CD4+, and CD8+ were determined using commercial test kits (Nanjing Jiancheng Bioengineering Institute, Nanjing, China), according to the instructions of the manufacturer (absorption spectrophotometry; iMark, BIORAD, Hercules, CA, USA).

Blood routine indexes were determined using the ProCyte Dx Hematology Analyzer (IDEXX Laboratories, Westbrook, ME, USA), and blood biochemical indexes were determined with the Hitachi 7020 automatic biochemical analyzer (HITACHI, Kanto Area, Japan).

### 2.5. Intestinal and Organ Histology

Duodenum, jejunum, and ileum sections were collected, and the contents were gently extruded. The sections were fixed in 4% paraformaldehyde and embedded in paraffin, and then cut into 4 μm thick slices and stained with hematoxylin and eosin. The examination was performed using a light microscope (Nikon Eclipse Ci, Tokyo, Japan) and microphotography with a digital camera attached to the microscope (Nikon digital sight DS-FI2, Tokyo, Japan). The villus height and crypt depth were calculated by measuring 10 villi and 10 crypts on five fields of view at points where the villi were connected to the lumen [27]. A histopathological examination was performed by an experienced histopathologist.

The heart, liver, spleen, lungs, and kidney were removed, trimmed of any superficial fat or blood, blotted dry, and weighed. Liver and kidney samples were fixed in 4% paraformaldehyde and subjected to the same staining procedure as for the small intestine. A histopathological examination was performed by an experienced histopathologist.

### 2.6. Intestinal Health Indexes

Secretory immunoglobulin A (sIgA) in ileum mucosa was measured using commercially available swine ELISA kits (R&D Systems, Minneapolis, MN, USA). The final results were calculated in the form of a microgram of sIgA per gram of protein.

The total RNA extracted from the ileum tissues and cDNA synthesis were conducted using the HiPure Total RNA Mini Kit (Magen, Guangzhou, China) and PrimeScriptTM RT reagent Kit (Takara Biotechnology Co. Ltd., Otsu, Shiga, Japan), respectively. The mRNA levels of the individual genes were measured with a real-time PCR (Kit: TB GreenTM Premix Ex TaqTM II; Takara Biotechnology Co. Ltd., Otsu, Shiga, Japan. Instrument: LightCycler Real-Time PCR System; Roche, Germany). The primers for the real-time PCR are listed in Table 3. The relative mRNA expression of the immune indexes was determined using the 2^−∆∆Ct^ method.

### 2.7. Microbiota Composition by 16S rRNA Sequencing Analysis

Fecal samples were collected from pigs in the control, 250, 500, 750, and 1000 mg/kg groups (six/group) on d 14 and 28. After genomic DNA extraction (E.Z.N.A.^®^ soil DNA Kit; Omega Bio-tek, Norcross, GA, USA), the V3-V4 hypervariable region was amplified with primers 338F (5′-ACTCCTACGGGAGGCAGCAG-3′) and 806R (5′-GGACTACHVGGGTWTCTAAT-3′) and then sequenced (Illumina MiSeq platform; Illumina, San Diego, CA, USA). The Operational taxonomic Units (OTUs) were clustered with 97% similarity by UPARSE62 (version 7.1 http://drive5.com/uparse/, accessed on 8 January 2022). Random samples of all the sample effective sequences were obtained by taking the minimum number of effective sequences in the sample. The sequences were aligned with the Silva (SSU115) 16S rRNA database with an RDP classifier (http://rdp.cme.msu.edu/, accessed on 8 January 2022) with a confidence threshold of 70% [28].

### 2.8. Statistical Analysis

Statistical analysis was performed using Prism software (GraphPad 7.0). No samples or animals were excluded from the analyses. Differences among > 2 groups with only one variable were assessed using a one-way ANOVA with a Tukey post hoc test. A distance-based redundancy analysis (db-RDA) was performed using the Euclidean distance. Taxonomic comparisons were analyzed with a Kruskal–Wallis H test with an fdr post hoc test, and a spearman correlation analysis was performed between different taxa and diarrhea-associated parameters (R version 3.3.1, pheatmap package).

## 3. Results

### 3.1. Growth Performance and Diarrhea Incidence

The effects of dietary C7 supplementation on growth performance are shown in Table 4. At 0–14 d, the 500 mg/kg group had the highest ADG and ADFI (quadratic, *p* < 0.05). At 15–28 d and 0–28 d, the F:G of the 500 mg/kg group was significantly lower than that of the other groups (*p* < 0.05). At 0-28 d, the ADG of the 500 mg/kg group was significantly higher than that of the other groups (*p* < 0.05). The results showed that dietary supplementation with 500 mg/kg C7 increased the ADG during the whole trial period (0–28 d) and decreased F:G over the later period (15–28 d) and the whole period (0–28 d), and thus was helpful in improving the performance of weaned piglets.

As shown in Table 5, compared with the control group, high-dose C7 supplementation (5000 mg/kg) had no significant effect on ADG, ADFI, or F:G at any stage of the experiment (*p* > 0.05). However, at 0–14 d, ADG was significantly lower in the 5000 mg/kg group than the 500 mg/kg group, and at every stage, F:G was higher in the 5000 mg/kg group than the 500 mg/kg group.

In terms of the diarrhea index (Table 6), for groups given a dietary supplementation of 250, 500, 750, or 1000 mg/kg C7, the diarrhea index was significantly lower than that of the control group in the 0–7 d stage (*p* < 0.05), and the index decreased linearly with an increasing dose (linear, *p* < 0.05). Dietary supplementation with 250, 500, and 750 mg/kg C7 resulted in a significantly lower diarrhea index than the control diet at the 0–28 d stage (*p* < 0.05), and the 500 mg/kg group had the lowest index (linear and quadratic, *p* < 0.05). Diarrhea frequency (Table 6) in pigs given a dietary supplementation with 250, 500, and 750 mg/kg C7 was significantly lower than that in the control group at the 0–28 d stage (*p* < 0.05), and the 500 mg/kg group had the lowest diarrhea frequency (linear, *p* < 0.05).

### 3.2. Apparent Total Tract Digestibility of Nutrients

As shown in Table 7, CP digestibility and energy digestibility were the highest in the 500 mg/kg group (linear, *p* < 0.05), but there was no significant difference between the treatment groups (*p* > 0.05). Compared with the control group, the dietary supplementation with 250 and 500 mg/kg C7 significantly increased EE digestibility (*p* < 0.05), and dietary supplementation with 250, 500, and 750 mg/kg C7 significantly increased Ca digestibility (*p* < 0.05).

### 3.3. Blood Indexes

The effects of dietary supplementation with different concentrations of C7 on the blood immune indexes are shown in Table 8. Compared with the control group, a 250 and 500 mg/kg C7 supplementation significantly increased IgG levels at d 14 (*p* < 0.05). At d 28, supplementation with 250 mg/kg C7 significantly increased IgG levels (*p* < 0.05), while 250 and 500 mg/kg C7 significantly decreased TNF-α levels (*p* < 0.05).

As shown in Table 9, the high dose of C7 (5000 mg/kg) had no significant effects on various routine blood indicators (*p* > 0.05). The effects of high-dose C7 on blood biochemical indexes are shown in Table 10. The results showed that the blood glucose (GLU) of the 5000 mg/kg group was significantly higher than that of the control group at d 14 (*p* < 0.05), but there were no significant differences in other indexes among groups (*p* > 0.05).

### 3.4. Small Intestinal Health Index

As shown in Table 11, compared with the control group, a diet supplemented with 500 mg/kg C7 tended to decrease duodenal crypt depth (*p* < 0.10) and significantly increase the ratio of the duodenal villus height to crypt depth (*p* < 0.05), ileum villus height (*p* < 0.05), and the ratio of the ileum villus height to crypt depth (*p* < 0.05).

The effects of dietary supplementation with different levels of C7 on ileal tight junction protein expression in weaned piglets are shown in Table 11. The results showed that the mRNA expressions of ZO-1 and Claudin-1 in the 500 mg/kg group and 1000 mg/kg group was significantly higher than that in the control group (*p* < 0.05).

In terms of intestinal immune performance (Table 11), compared with the control group, ileal IL-8 expression was significantly increased in the 1000 mg/kg group and IL-10 expression was significantly increased in both the 500 mg/kg group and 1000 mg/kg group.

### 3.5. Morphological Characteristics and Organ Weights

The effects of the graded levels of C7 supplementation on small intestinal, liver, and kidney morphology are shown in Figure 1. In each treatment group, the intestinal villus epithelial cells were intact, the structure of the intestinal lamina propria was normal, and there was no edema or other lesions. In addition, tissue morphology and structure were normal, without obvious pathological changes. The structure of the hepatic lobule was clear, the hepatic cords were arranged neatly with normal morphology and structure, and no obvious pathological changes were observed. The renal tubular epithelial cells had a normal morphology and were closely arranged. No obvious pathological changes were observed in the kidneys.

Compared with the control group, C7 supplementation did not affect organ weight (*p* > 0.05, Table 12).

### 3.6. Microbiota Diversity Analysis

We analyzed 16s rRNA gene pyrosequencing data from 60 fecal samples (6/group from control, 250, 500, 750, and 1000 mg/kg groups on d 14 and d 28). No differences among community evenness and richness diversity, as assessed by the Shannon and Chao indexes, were noted across the control and C7 supplementation groups (Figure 2A,B).

As shown in Figure 2(C1), PCoA revealed the distinct microbial structures of the control and C7 supplementation groups on d 14 (*p* < 0.05). However, no obvious separation was observed among the treatment groups on d 28 (Figure 2(C2)). In addition, the results showed that the samples in the 500 mg/kg group exhibited lesser distances, indicating that the microbial community structures may have been more similar and stable (Figure 2C).

### 3.7. Correlation Analysis between Microbiota and Diarrhea Incidence

To further explore the role of the altered microbiota structure, db-RDA analysis was used. The results showed that on d 14, the microbiota structure of the control group was positively correlated with diarrhea incidence, while those of the C7 supplementation groups were negatively correlated with diarrhea incidence (Figure 3A). However, no such pattern was found on d 28 (Figure 3B).

### 3.8. Correlation Analysis between Specific Microbiota and Diarrhea Incidence

The most differential microorganisms between the five groups are shown in Figure 4A. After applying Spearman correlation analysis (Figure 4B), we found that *norank_f_Selenomonadaceae* and *Christensenellaceae* were significantly and negatively associated with diarrhea incidence, and their abundance significantly increased in the C7-treated groups (Figure 4(A1,B1)). The results for d 28 showed that *Monoglobus* was significantly and negatively associated with diarrhea incidence, and its abundance was significantly increased in the C7-treated groups, while *Collinsella* was significantly and positively associated with diarrhea incidence, and its abundance was significantly decreased in the C7-treated groups (Figure 4(A2,B2)).

## 4. Discussion

Diarrhea in weaned piglets has been one of the most perplexing problems in the pig industry. Especially after the comprehensive ban of antibiotic growth promoters, the diarrhea problem became more prominent. The increased incidence of diarrhea, increased postweaning mortality, and prolonged slaughter time have led to an urgent need for new, safe, and effective alternatives to growth promoters. Consequently, AMPs have received attention due to their broad-spectrum activity, speed of action, and low propensity to induce resistance in bacteria [9,14,29,30]. C7 is an antimicrobial peptide with bactericidal and immunomodulatory activity [20,31]. The present study evaluated the efficacy and safety of C7 as a potential feed additive for weaned piglets.

Piglets fed a diet containing 500 mg/kg C7 had an improved ADG at 0–28 d and decreased F:G (15–28 d and 0–28d), which suggested the ability of C7 to improve the growth performance of weaned piglets. These findings are in agreement with previous studies that reported improvements in the ADG and feed efficiency of weanling pigs fed diets supplemented with antimicrobial peptides [15,16,18,32]. When the amount of C7 added reached 5000 mg/kg, growth performance was not adversely affected. In the present study, the supplementation of weanling piglet diets with 500 mg/kg C7 led to a greater ATTD of EE and Ca, which could partly explain the results in growth performance. The improved ATTD of nutrients with C7 supplementation might have been due to the increased availability of nutrients for intestinal absorption and the alteration in intestinal morphology, intestinal epithelium thickness, and epithelial cell turnover [33,34,35]. Consistent with this, the results showed that C7 improved the intestinal morphology of the duodenum and ileum to a certain extent and increased the expression level of ileal tight junction proteins, which are beneficial to epithelial function and nutrient absorption in weaned piglets [36,37,38].

Weaned piglets’ immune systems begin developing at approximately 3 weeks of age, but they are unable to mount an effective immune response until around 5 weeks of age [39]. However, piglets usually wean when they reach 4- to 5-weeks-old, at which point there is an increase in the concentration of serum immunoglobulin and other nonspecific immunity factors required to regulate and enhance immune functions, which provides health benefits, diminishes weaning stress, and improves the health status and growth performance of weanling pigs [40,41]. Our results indicate that C7 supplementation improved serum IgG and anti-inflammatory cytokine IL-10 levels and decreased proinflammatory cytokine TNF-α levels, which is consistent with previous reports that antimicrobial peptides improved the growth performance and immunity of piglets [19,30]. Thus, supplementation with C7 improved some immune indicators, and this improvement may help alleviate the stress-induced negative state.

Postweaning diarrhea (PWD) is an economically important disease in pig that leads to financial losses as a result of profuse diarrhea, dehydration, reduced performance, and medication expenses [42,43,44]. In the present study, a dietary supplementation with C7 significantly alleviated incidences of diarrhea in weanling pigs, and diarrhea incidence was lowest when the supplemental level was 500 mg/kg. Based on the above results, the recommended amount of C7 to use as an additive is 500 mg/kg. Although high doses of C7 significantly raised blood GLU levels, they were still within the normal range [45]. No other intestinal or organ abnormalities were observed, and all serum biochemistry and most hematological parameters were within normal ranges, indicating the safety of C7.

The gut is a common site for nutrient digestion, microbial community colonization, and immune cell localization, and this geographical proximity largely determines their interactions. Therefore, we performed a fecal microbiota analysis. One of the novel aspects of the present study was that diarrhea incidence was correlated with microbiota structure, as shown by our comprehensive analysis, and the evidence suggested that C7 promotes gut health by selectively regulating specific microbial taxa. Although there was no significant difference in microbial diversity among the groups, the beta diversity provided evidence for differences in the composition of microbial communities in the C7 supplemented groups, especially on d 14. In the present study, the significant relief of diarrhea symptoms in the C7 supplemented groups suggested a key role for C7-regulated microbiota in alleviating PWD. Similarly, many other studies have shown that antimicrobial peptides beneficially affect host animals by improving microbial homeostasis and creating microecological conditions that inhibit harmful microorganisms and favor beneficial ones [33,34,46,47,48]. However, these studies only described changes in the microbiota and did not delve into potential associations. The present experiments provided preliminary evidence that C7-induced alterations to microbiota can alleviate diarrhea, and the key changes induced by C7, i.e., blooms of *norank_f_Selenomonadaceae*, *Christensenellaceae,* and *Monoglobus* combined with a reduction in *Collinsella*, were closely related to the recovery of diarrhea. Given the fact that the way in which C7 regulates microbiota is still not fully understood, the results presented here provide experimental evidence that C7 is a possible treatment for PWD and a source of information for future microbiome research. The microbial structural changes in the intestine of weanling piglets reported in the present study provide new information on the potential use of C7 as an alternative to antibiotic growth promoters in pigs.

## 5. Conclusions

The results obtained in the present study indicate that Microcin C7 has beneficial effects on the growth performance, diarrhea incidence, ATTD of EE and Ca, immune property, intestinal morphology, and microbiome structure of piglets. An association analysis showed that the diarrhea-relieving effect of C7 was significantly correlated with its regulatory effect on specific microorganisms. Under the conditions of the present study, the most appropriate supplemental level was 500 mg/kg; no intestinal or organ abnormalities were observed in the high-dose group, and all serum biochemical and most hematological parameters were within normal ranges. In conclusion, our results suggest that C7 is safe and effective as a potential alternative to antibiotic growth promoters in weaned piglets. However, the way in which C7 regulates the microbiota and the mode of action are still not fully understood, so additional studies on the mechanisms of microbiota regulation and diarrhea resolution should help to clarify this question.

## Figures and Tables

**Figure 1 animals-12-03267-f001:**
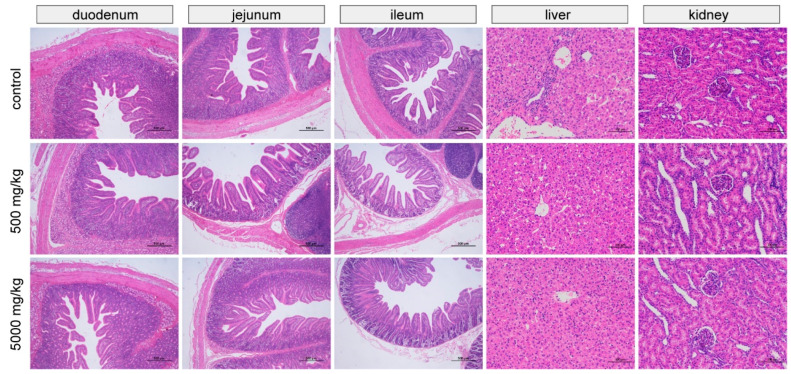
Representative images of small intestinal, liver, and kidney by H&E staining.

**Figure 2 animals-12-03267-f002:**
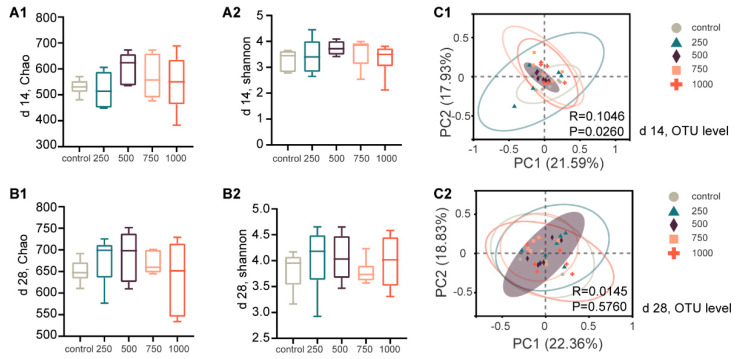
Microbiota changes in diversity and structure. (**A**,**B**) Alpha diversity comparisons of microbial communities (d 14: **A1**,**A2**; d 28: **B1**,**B2**). (**C1**,**C2**) PCoA of 16s genes. Using an OTU definition of 97% similarity, based on bray_curtis distance algorithm, significance test was carried out using PERMANOVA.

**Figure 3 animals-12-03267-f003:**
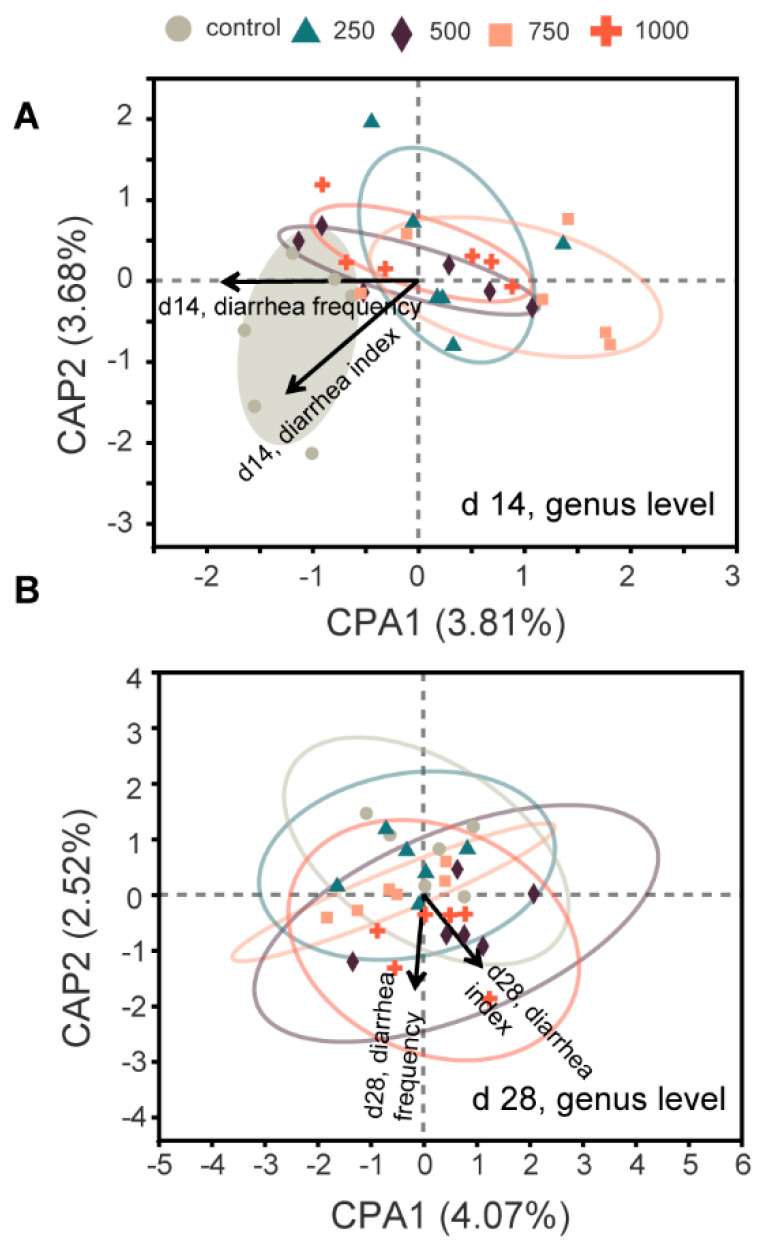
Distance-based redundancy analysis between microbiota and diarrhea incidence. (**A**), d 14; (**B**), d 28.

**Figure 4 animals-12-03267-f004:**
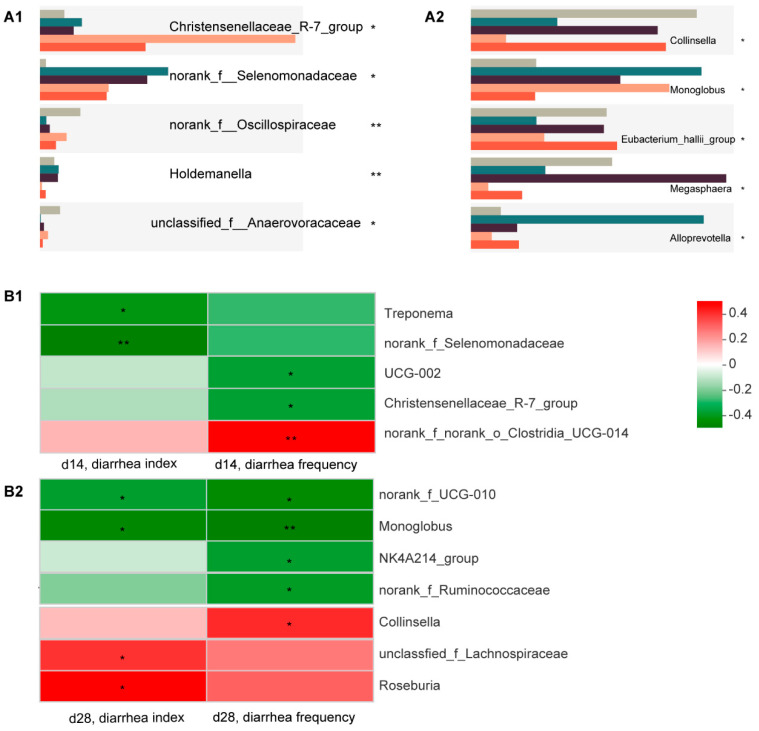
Bacterial targets of Microcin C7. (**A**) Relative abundances of top 5 genera at d 14 (**left**) and d 28 (**right**). An asterisk indicates significant between-group differences (* *p* < 0.05; ** *p* < 0.01). (**B**) Correlation matrix between specific taxa and diarrhea-associated parameters at d 14 (**up**) and d 28 (**down**). An asterisk indicates that the correlation is statistically significant (* *p* < 0.05; ** *p* < 0.01).

**Table 1 animals-12-03267-t001:** Ingredient composition and nutrient content of the basal diet (as-fed basis, %).

Items	0–14 d	15–28 d
Corn	59.52	63.85
Soybean meal	7.00	6.30
Extruded full-fat soybean	4.40	4.00
Soy protein concentration	8.00	5.90
Fish meal	7.20	7.00
Whey powder	10.20	9.80
Soybean oil	1.30	0.74
Dicalcium phosphate	0.55	0.55
Limestone	0.80	0.60
Sodium chloride	0.30	0.30
L-Lysine-HCl	0.16	0.16
DL-Methionine	0.05	0.03
L-Tryptophan	0.02	0.02
Chromic oxide	-	0.25
Permix ^1^	0.50	0.50
Nutrient levels ^2^		
DE (Mcal/kg)	3.591	3.538
CP	20.58	18.93
Ca	0.84	0.75
Total P	0.66	0.63
Lys	1.35	1.23
Met	0.41	0.39
Thr	0.81	0.74
Trp	0.24	0.21

DE, digestible energy. ^1^ Vitamin–mineral premix provided the following per kg of complete diet: 0–14 d: vitamin A, 14.0 KIU; vitamin D3, 6.0 KIU; vitamin E, 30.0 IU; vitamin K3, 2.5 mg; vitamin B1, 2.5 mg; vitamin B2, 4.0 mg; vitamin B6, 3.0 mg; vitamin B12, 20.0 μg; niacinamide, 40.0 mg; pantothenic acid, 12.5 mg; folacin, 0.7 mg; biotin, 70.00 μg; Fe, 150.0 mg; Cu, 125.0 mg; Zn, 90.0 mg; Mn, 65.0 mg; I, 0.625 mg; Se, 0.325 mg. 15–28 d: vitamin A, 11.0 KIU; vitamin D3, 3.25 KIU; vitamin E, 15.0 IU; vitamin K3, 1.5 mg; vitamin B1, 0.8 mg; vitamin B2, 3.0 mg; vitamin B6, 1.5 mg; vitamin B12, 10.0 μg; niacinamide, 18.0 mg; pantothenic acid, 9.5 mg; folacin, 0.4 mg; biotin, 25.00 μg; Fe, 125.0 mg; Cu, 100.0 mg; Zn, 90.0 mg; Mn, 50.0 mg; I, 0.775 mg; Se, 0.35 mg. ^2^ All nutrient levels except digestible energy were analyzed and values are the means of two determinations. Piglets had free access to feed and water. On the last day of the experiment, pigs (six/group) in the control, 500 mg/kg, 1000 mg/kg, and 5000 mg/kg groups with weights close to the average weight of the replicates were selected and humanely killed by exsanguination after electrical stunning to obtain samples.

**Table 2 animals-12-03267-t002:** Scoring system for calculating diarrhea ^1^.

Degree of Diarrhea	Stool Consistency	Score
Normal	Formed	0
Mild	Soft but still formed	1
Moderate	Soft	2
Severe	Very soft, wet	3

^1^ The feces of piglets were observed twice a day and scored, and the highest score of each pig was used as the diarrhea index of the day.

**Table 3 animals-12-03267-t003:** Primers for real-time PCR.

Gene	Primer Sequence (5′-3′)
IL-1β	F: ACTCATTGTGGCTGTGGAGA
	R: TTGTTCATCTCGGAGCCTGT
TNF-α	F: ACCCTCACACTCACAAACCA
	R: GGCAGAGAGGAGGTTGACTT
IL-8	F: ATGACTTCCAAACTGGCT
	R: GGTCCACTCTCAATCACT
IL-10	F: CGACTCAACGAAGAAGGCACAG
	R: CTCTGACAAGGCTTGGCAACC
SIgA	F: CCTCCACCAGCTCATACCCTG
	R: GGTGAAAATCCCATTCCGAGT
MUC1	F: AGGCCAGGATCTGTACTGGTAGAG
	R: TGGTAGGTGGGGTACTCGCTCATA
ZO-1	F: AAGGATGTTTACCGTCGCATT
	R: ATTGGACACTGGCTAACTGCT
Occludin	F: GTGGTAACTTGGAGGCGTCTTC
	R: CCGTCGTGTAGTCTGTCTCGTA
Claudin-1	F: GCTGGGTTTCATCCTGGCTTCT
	R: CCTGAGCGGTCACGATGTTGTC
β-actin	F: TAGGCGGACTGTTACTGAGC
	R: GCCTTCACCGTTCCAGTTTT

**Table 4 animals-12-03267-t004:** Effects of dietary C7 supplementation on growth performance.

Item	C7 (mg/kg Diet)					SEM	*p*-Value		
	0	250	500	750	1000		ANOVA	Linear	Quadratic
Body weight									
0 d, kg	7.90	7.91	7.91	7.90	7.91	0.03	0.99	0.89	0.69
14 d, kg	13.27	13.66	13.94	13.30	13.42	0.18	0.08	0.30	0.03
28 d, kg	20.53^b^	20.67^b^	21.43^a^	20.41^b^	20.49^b^	0.23	<0.01	0.35	0.13
0–14 d									
**ADG, kg**	**0.38**	**0.41**	**0.43**	**0.39**	**0.39**	**0.01**	**0.08**	**0.33**	**<0.05**
**ADFI, kg**	**0.59**	**0.61**	**0.62**	**0.57**	**0.59**	**0.01**	**0.11**	**0.74**	**0.03**
F:G	1.54	1.49	1.42	1.48	1.51	0.05	0.46	0.16	0.46
15–28 d									
ADG, kg	0.52	0.50	0.53	0.51	0.51	0.01	0.17	0.89	0.70
ADFI, kg	0.85	0.83	0.80	0.85	0.86	0.02	0.30	0.39	0.34
**F:G**	**1.64 ^a^**	**1.64 ^a^**	**1.50 ^b^**	**1.67 ^a^**	**1.71 ^a^**	**0.04**	**0.04**	**0.31**	**0.44**
0–28 d									
**ADG, kg**	**0.45 ^b^**	**0.46 ^b^**	**0.48 ^a^**	**0.45 ^b^**	**0.45 ^b^**	**0.01**	**<0.05**	**0.36**	**0.16**
ADFI, kg	0.72	0.72	0.71	0.71	0.73	0.01	0.90	0.44	0.82
**F:G**	**1.60 ^a^**	**1.57 ^a^**	**1.46 ^b^**	**1.59 ^a^**	**1.62 ^a^**	**0.03**	**0.02**	**0.09**	**0.23**

ADG, average daily weight gain; ADFI, average daily feed intake; F:G, feed-to-gain ratio. ^a, b^ Means in the same row with different superscripts differ (*p* < 0.05). The bold means to emphasize indicators with significant differences.

**Table 5 animals-12-03267-t005:** Effects of high-dose C7 supplementation on growth performance of weaned piglets.

Item	C7 (mg/kg Diet)			SEM	*p*-Value
	0	500	5000		
Body weight					
0 d, kg	7.90	7.91	7.90	0.13	0.99
14 d, kg	13.27	13.94	13.16	0.21	0.27
28 d, kg	20.53	21.43	20.45	0.29	0.32
0–14 d					
**ADG, kg**	**0.38 ^b^**	**0.43 ^a^**	**0.38 ^b^**	**0.02**	**0.04**
ADFI, kg	0.59	0.62	0.56	0.03	0.63
**F:G**	**1.54 ^b^**	**1.42 ^a^**	**1.49 ^b^**	**0.04**	**0.01**
15–28 d					
ADG, kg	0.52	0.53	0.52	0.02	0.09
ADFI, kg	0.85	0.80	0.86	0.04	0.63
**F:G**	**1.64 ^a^**	**1.50 ^b^**	**1.66 ^a^**	**0.06**	**<0.01**
0–28 d					
ADG, kg	0.45	0.48	0.45	0.02	0.08
ADFI, kg	0.72	0.71	0.71	0.04	0.99
**F:G**	**1.60 ^a^**	**1.46 ^b^**	**1.60 ^a^**	**0.04**	**<0.01**

ADG, average daily weight gain; ADFI, average daily feed intake; F:G, feed-to-gain ratio. ^a, b^ Means in the same row with different superscripts differ (*p* < 0.05). The bold means to emphasize indicators with significant differences.

**Table 6 animals-12-03267-t006:** Effects of dietary C7 supplementation on the diarrhea incidence of weaned piglets (%).

Item	C7 (mg/kg Diet)					SEM	*p*-Value		
	0	250	500	750	1000		ANOVA	Linear	Quadratic
Diarrhea index									
**0–7 d**	**8.33 ^a^**	**3.97 ^b^**	**3.97 ^b^**	**3.18 ^b^**	**4.37 ^b^**	**1.06**	**0.02**	**<0.01**	**0.11**
8–14 d	2.38	1.19	1.59	1.59	1.59	0.71	0.83	0.49	0.38
0–14 d	5.36	2.58	2.78	2.38	2.98	0.76	0.07	0.02	0.12
15–28 d	6.54	3.57	1.98	4.84	6.15	1.13	0.06	0.04	0.10
**0–28 d**	**5.95 ^a^**	**3.08 ^b^**	**2.38 ^b^**	**3.57 ^b^**	**4.56 ^ab^**	**0.72**	**0.02**	**<0.01**	**0.04**
Diarrhea frequency									
0–7 d	13.49	6.35	7.54	4.37	8.33	2.49	0.15	0.03	0.45
8–14 d	2.78	1.59	1.59	1.19	1.59	0.78	0.67	0.21	0.59
0–14 d	8.14	3.97	4.56	2.78	4.96	1.47	0.16	0.03	0.44
15–28 d	18.85	8.33	6.15	11.07	17.46	3.28	0.05	0.02	0.13
**0–28 d**	**13.49 ^a^**	**6.15 ^b^**	**5.36 ^b^**	**6.85 ^b^**	**11.21 ^ab^**	**2.02**	**0.04**	**<0.01**	**0.13**

^a, b^ Means in the same row with different superscripts differ (*p* < 0.05). The bold means to emphasize indicators with significant differences.

**Table 7 animals-12-03267-t007:** Effects of dietary C7 supplementation on nutrients’ apparent total tract digestibility (%).

Item	C7 (mg/kg Diet)					SEM	*p*-Value		
	0	250	500	750	1000		ANOVA	Linear	Quadratic
**CP**	82.01	83.05	83.61	83.44	83.10	0.48	0.21	0.04	0.32
GE	86.68	87.75	87.78	87.33	87.21	0.28	0.10	0.11	0.04
**EE**	**67.63 ^c^**	**72.73 ^a^**	**70.61 ^ab^**	**67.69 ^c^**	**68.62 ^bc^**	**0.60**	**<0.01**	**0.33**	**<0.01**
OM	89.09	89.78	89.82	89.52	89.53	0.24	0.32	0.21	0.12
**Ca**	**65.10 ^b^**	**72.73 ^a^**	**74.57 ^a^**	**74.07 ^a^**	**68.34 ^b^**	**1.06**	**<0.01**	**<0.01**	**0.02**
P	62.76	62.81	66.19	63.22	64.73	0.94	0.10	0.48	0.25
DM	88.55	89.19	89.21	88.88	88.94	0.24	0.37	0.29	0.12

CP, crude protein; GE, gross energy; EE, ether extract; OM, organic matter; DM, dry matter. ^a, b, c^ Means in the same row with different superscripts differ (*p* < 0.05). The bold means to emphasize indicators with significant differences.

**Table 8 animals-12-03267-t008:** Effect of dietary C7 supplementation on serum immune indexes.

Item	C7 (mg/kg Diet)					SEM	*p*-Value		
	0	250	500	750	1000		ANOVA	Linear	Quadratic
14 d									
IgA (g/L)	0.87	1.07	1.01	1.00	1.09	0.06	0.07	0.27	0.02
**IgG (g/L)**	**5.46 ^b^**	**6.82 ^a^**	**5.90 ^ab^**	**6.19 ^a^**	**5.97 ^ab^**	**0.26**	**0.02**	**0.10**	**0.03**
IgM (g/L)	0.67	0.78	0.71	0.74	0.74	0.03	0.22	0.36	0.12
IL-1β (pg/mL)	21.60	19.00	16.58	18.15	16.96	1.55	0.20	0.38	0.10
TNF-α (pg/mL)	157.80	134.10	112.30	129.90	124.50	10.28	0.06	0.15	0.02
IFN-γ (pg/mL)	89.06	79.31	72.98	70.37	69.09	5.57	0.11	0.07	0.39
IL-6 (pg/mL)	70.18	75.60	76.43	73.11	76.91	1.85	0.10	0.06	0.72
IL-10 (pg/mL)	25.49	28.21	29.91	33.90	33.54	2.02	0.09	0.05	0.89
CD3+ (ug/L)	11.20	11.08	10.37	10.64	10.88	0.31	0.36	0.11	0.80
CD4+ (ug/L)	6.83	6.25	6.87	6.63	6.59	0.27	0.53	0.89	0.35
CD8+ (ug/L)	4.69	4.13	4.13	4.21	4.18	0.16	0.12	0.05	0.12
CD4+/CD8+	1.45	1.52	1.64	1.61	1.58	0.06	0.20	0.06	0.53
28 d									
IgA (g/L)	1.07	1.09	1.09	1.03	1.07	0.03	0.69	0.50	0.28
**IgG (g/L)**	**6.69 ^b^**	**7.80 ^a^**	**6.85 ^ab^**	**6.80 ^ab^**	**5.97 ^b^**	**0.32**	**0.01**	**0.29**	**0.25**
IgM (g/L)	0.79	0.83	0.82	0.83	0.80	0.02	0.29	0.08	0.66
IL-1β (pg/mL)	19.97	21.01	17.94	19.48	19.46	1.43	0.66	0.65	0.19
**TNF-α (pg/mL)**	**140.62 ^a^**	**116.91 ^b^**	**119.63 ^b^**	**154.71 ^a^**	**137.20 ^ab^**	**8.95**	**0.04**	**0.72**	**0.01**
IFN-γ (pg/mL)	63.85	56.40	52.63	59.40	56.90	3.49	0.26	0.24	0.08
IL-6 (pg/mL)	80.52	74.7	70.46	72.59	74.21	2.85	0.18	0.04	0.27
IL-10 (pg/mL)	27.94	33.82	36.27	29.29	30.67	2.31	0.11	0.27	0.03
CD3+ (ug/L)	11.23	11.25	11.24	10.98	12.04	0.46	0.57	0.39	0.44
CD4+ (ug/L)	6.23	6.19	6.32	6.49	6.54	0.31	0.90	0.75	0.86
CD8+ (ug/L)	4.11	3.97	3.80	4.05	3.69	0.13	0.17	0.99	0.09
CD4+/CD8+	1.54	1.56	1.66	1.61	1.77	0.07	0.21	0.96	0.34

^a, b^ Means in the same row with different superscripts differ (*p* < 0.05). The bold means to emphasize indicators with significant differences.

**Table 9 animals-12-03267-t009:** Effects of high-dose C7 supplementation on blood routine indexes.

Item	C7 (mg/kg Diet)			SEM	*p*-Value
	0	500	5000		
**14 d**					
PLT, 10^9^/L	498.67	694.00	635.17	74.41	0.18
WBC, 10^9^/L	27.40	28.24	31.24	2.20	0.46
RBC, 10^9^/L	5.76	5.81	5.97	0.17	0.71
HGB, g/L	97.00	92.40	96.80	3.11	0.53
HCT	0.28	0.28	0.28	0.01	0.85
W-SCC, 10^9^/L	13.40	13.82	15.51	1.99	0.46
W-MCC, 10^9^/L	2.68	2.57	3.09	0.39	0.92
W-LCC, 10^9^/L	10.03	12.22	13.60	1.78	0.55
%L	51.03	46.43	47.87	3.23	0.45
%M	20.13	12.15	11.25	3.88	0.68
%G	39.33	41.43	40.88	2.87	0.58
MCV	49.00	47.42	46.92	0.76	0.11
MCH, f/L	16.80	16.17	16.68	0.30	0.19
MCHC	338.17	327.83	318.33	23.25	0.55
RDW-SD	19.50	21.10	20.05	0.76	0.38
RDW-CV	25.08	26.02	26.08	0.96	0.66
PCT	0.40	0.49	0.49	0.05	0.17
MPV	8.05	8.02	7.99	0.13	0.88
PDW	19.15	17.25	18.35	1.22	0.32
**Item**	**C7 (mg/kg Diet)**			**SEM**	***p*-Value**
	**0**	**500**	**5000**		
**28 d**					
PLT, 10^9^/L	789.70	991.00	780.80	99.12	0.63
WBC, 10^9^/L	31.29	29.76	32.69	2.03	0.72
RBC, 10^9^/L	6.23	6.35	6.07	0.09	0.21
HGB, g/L	102.00	103.80	100.40	2.64	0.67
HCT	0.27	0.28	0.27	0.01	0.82
W-SCC, 10^9^/L	12.03	14.14	14.49	1.23	0.35
W-MCC, 10^9^/L	2.88	3.24	3.51	0.52	0.68
W-LCC, 10^9^/L	14.85	14.86	14.95	1.51	0.82
%L	41.95	42.47	45.07	2.44	0.46
%M	9.40	10.58	10.43	0.97	0.52
%G	48.65	46.95	44.50	2.22	0.75
MCV	44.25	44.23	44.58	0.92	0.66
MCH, f/L	16.43	16.48	16.43	0.21	0.79
MCHC	352.83	354.83	349.67	4.02	0.94
RDW-SD	20.50	20.78	21.03	0.22	0.58
RDW-CV	23.62	24.23	25.57	0.62	0.18
PCT	0.59	0.61	0.59	0.05	0.78
MPV	7.45	7.47	7.50	0.06	0.64
PDW	12.77	13.58	14.38	1.78	0.68

PLT, blood platelet; WBC, white blood cell count; RBC, red blood cell; HGB, hemoglobin; HCT, hematocrit; W-SCC, absolute value of lymphocytes; W-MCC, absolute value of intermediate cell; W-LCC, absolute neutrophil value; %L, percentage of lymphocytes; %M, percentage of intermediate cell; %G, percentage of granulocyte; MCV, mean corpuscular volume; MCH, mean corpuscular hemoglobin; MCHC, mean corpuscular hemoglobin concentration; RDW, red blood cell; SD, standard deviation; CV, coefficient of variation; PCT, plateletcrit; MPV, mean platelet volume; PDW, platelet distribution width.

**Table 10 animals-12-03267-t010:** Effect of high-dose C7 supplementation on serum biochemistry parameters.

Item	C7 (mg/kg diet)			SEM	*p*-Value
	0	500	5000		
14 d					
AST, U/L	63.65	54.80	64.04	6.19	0.09
ALT, U/L	57.28	53.28	62.82	7.29	0.72
ALP, U/L	387.57	388.38	368.48	31.67	0.89
TP, g/L	50.68	49.90	52.60	1.25	0.43
ALB, g/L	21.65	22.53	22.83	0.59	0.39
**GLU, mmol/L**	**5.13 ^b^**	**5.62 ^ab^**	**6.09 ^a^**	**0.15**	**0.01**
Urea, mmol/L	1.08	1.19	1.31	0.15	0.60
CREA, μmol/L	88.55	87.85	93.62	4.46	0.64
TBIL, umol/L	7.77	6.69	6.93	0.69	0.58
28 d					
AST, U/L	37.27	47.08	36.92	3.62	0.14
ALT, U/L	45.90	47.85	43.75	2.25	0.53
ALP, U/L	281.67	259.72	286.27	23.71	0.71
TP, g/L	57.78	54.90	54.28	1.61	0.31
ALB, g/L	24.35	24.50	22.62	0.80	0.23
GLU, mmol/L	5.76	6.21	5.83	0.17	0.93
Urea, mmol/L	0.58	0.63	0.64	0.05	0.43
CREA, μmol/L	80.28	84.23	84.10	3.43	0.66
TBIL, umol/L	6.12	5.92	5.86	0.68	0.97

AST, aspartate transaminase; ALT, alanine transaminase; ALP, alkaline phosphatase; TP, total protein; ALB, albumin; GLU, blood glucose; CREA, creatinine; TBIL, total bilirubin. ^a, b^ Means in the same row with different superscripts differ (*p* < 0.05). The bold means to emphasize indicators with significant differences.

**Table 11 animals-12-03267-t011:** Effect of dietary C7 supplementation on intestinal health indexes.

Items	C7 (mg/kg Diet)			SEM	*p*-Value
	0	500	1000		
Villus height, μm					
Duodenum	467.6	521.1	501.7	19.0	0.14
Jejunum	503.0	518.5	506.8	16.0	0.79
**Ileum**	**388.4 ^b^**	**447.0 ^a^**	**420.6 ^ab^**	**12.9**	**0.01**
Crypt depth, μm					
**Duodenum**	**262.3 ^a^**	**229.2 ^b^**	**246.1 ^ab^**	**10.5**	**0.09**
Jejunum	254.9	245.3	247.6	8.9	0.74
Ileum	228.5	208.8	211.9	9.6	0.30
Villus height/crypt depth					
**Duodenum**	**1.8 ^b^**	**2.4 ^a^**	**2.1 ^ab^**	**0.1**	**0.01**
Jejunum	2.0	2.2	2.1	0.1	0.58
**Ileum**	**1.8 ^b^**	**2.3 ^a^**	**2.1 ^ab^**	**0.1**	**0.02**
mRNA expression in ileum					
MUC-1	1.00	1.16	1.13	0.21	0.86
**ZO-1**	**1.00 ^b^**	**2.54 ^a^**	**1.82 ^ab^**	**0.36**	**0.03**
Occludin	1.00	3.25	1.29	0.60	0.04
**Claudin-1**	**1.00 ^b^**	**1.96 ^a^**	**1.13 ^b^**	**0.18**	**0.01**
IL-1β	1.00	0.72	0.75	0.48	0.61
**IL-8**	**1.00 ^b^**	**1.30 ^ab^**	**2.16 ^a^**	**0.58**	**0.03**
IL-10	1.00	1.53	1.32	0.27	0.06
TNF-α	1.00	0.81	1.05	0.45	0.65
sIgA in ileum mucosa	0.39	0.49	0.50	0.07	0.08

^a, b^ Means in the same row with different superscripts differ (*p* < 0.05). The bold means to emphasize indicators with significant differences.

**Table 12 animals-12-03267-t012:** Effect of high-dose C7 supplementation on organ weights (% BW) in weaned piglets.

Item	C7 (mg/kg Diet)			SEM	*p*-Value
	0	500	5000		
Heart	5.54	5.55	5.74	0.26	0.84
Liver	30.45	29.33	31.02	1.19	0.47
Spleen	2.00	1.88	2.07	0.11	0.52
Lung	11.05	10.78	10.97	0.42	0.90
Kidney	2.87	2.60	2.79	0.11	0.27

## Data Availability

The data presented in this study are available from the corresponding author upon request.

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
