# Peer review of "Evaluation of Effectiveness and Safety of Microcin C7 in Weaned Piglets"

_animals, 2022, doi:10.3390/ani12233267_

Round 1
Reviewer 1 Report
This manuscript tested the effect and safety of microcin C7. The experiment was well designed and the manuscript was well-written. Minor revisions are needed:
1. Suggest deleting “graded levels of” in the title.
2. It needs to note that the microcin C7 used in the experiment is made in the lab or purchased from the company.
3. Table 4 and Table 5 show the body weight at day 0 equal to or more than 7.90 kg, however, line 73 shows the average body weight is 7.86 kg, please double check it.
4. It is better to use the chi-square test to do the statistical analysis for diarrhea frequency.
Author Response
We would like to thank the reviewers for your professional comments and suggestions, which are very helpful in improving our manuscript. Based on the instructions provided in your letter, we uploaded the file of the revised manuscript. Appended to this letter is our point-by-point response to the comments raised by the reviewers.
Reviewer 1
Suggestions 1. Suggest deleting “graded levels of” in the title.
Suggestions 2. It needs to note that the microcin C7 used in the experiment is made in the lab or purchased from the company.
Reply: We feel sorry for the vague statement. Follow your suggestion, we have revised the above sections.
Suggestions 3. Table 4 and Table 5 show the body weight at day 0 equal to or more than 7.90 kg, however, line 73 shows the average body weight is 7.86 kg, please double check it.
Reply: Thank you so much for your careful check. We didn't check it carefully and may have made some mistakes in uploading. Initial average body weight should be 7.90.
Suggestions 4. It is better to use the chi-square test to do the statistical analysis for diarrhea frequency.
Reply: We gratefully appreciate for your nice comment. We totally agree with you, comparisons between Nominal data, and data of frequency type, should be tested by chi-square. However, this experiment is a completely randomized design with multiple outcomes and multiple groups, which can also be applied to the rank-sum test. In addition, the results of chi-square test showed the same trend.
Reviewer 2 Report
The article “Evaluation of effectiveness and safety of graded levels of Mi-crocin C7 in weaned piglets” by Shang et al needs improvements. The introduction is week and should be improved considerably. In addition, authors must address the following comments:
Line 9: Please correct the word “m+echanisms”.
Line 100, 189 & 197: F:G was denoted as feed to gain ratio at line 100. However, the same was denoted as feed conversion ratio at line 189 and 197. Please use FCR for this and be consistent throughout the manuscript.
Lines 160-161: Please clarify why fecal samples were not collected from the group fed with diet supplemented with 5000mg/kg for microbiota analysis?
Tables: Lines 78 and 79 report that five additional diets were formulated by adding 250, 500, 750, 1000 or 5000 mg/kg C7. Whereas the tables in the results section do not show the effects of all the formulated diets. Some of the tables show the effects of diet supplemented with 250, 500, 750 and 1000 mg/kg supplementation of the additive but not the 5000 mg/kg (table 4, 6, 7 and 8). on the other hand, tables 5, 9, 10 shows the effects of diet supplemented with only two levels of the additive (500 and 5000 mg/kg) but not the others. Why the effects of all the experimental diets have not been reported in the results?
Lines 226, 228, 235 and 243: Please check numbering of the tables. Two tables have been numbered as table 8 and two tables have been numbered as table 9.
Lines 325-332: These lines are suitable for introduction section.
Lines 347-352: Please provide a context of these lines with reference to this study.
Discussion: This section is week. Please improve the section.
Lines 390-391 & 395-396. This information is repeating in simple summary and abstract. Please avoid repetition.
Conclusions: Conclusions should be improved and re-written in connection with the specific findings of the study.
Author Response
We would like to thank the reviewers for your professional comments and suggestions, which are very helpful in improving our manuscript. Based on the instructions provided in your letter, we uploaded the file of the revised manuscript. Appended to this letter is our point-by-point response to the comments raised by the reviewers.
Suggestions 1. Please correct the word “m+echanisms”.
Reply: Thank you so much for your careful check. We did not have this problem in the manuscript, there may have been some mistakes in the upload process. Thank you so much for your careful check.
Suggestions 2. Line 100, 189 & 197: F:G was denoted as feed to gain ratio at line 100. However, the same was denoted as feed conversion ratio at line 189 and 197. Please use FCR for this and be consistent throughout the manuscript.
Reply: Thank you for your nice advice. "feed to gain ratio" is correct. The following mistake caused readers' misunderstanding, and we have corrected it.
Suggestions 3. Lines 160-161: Please clarify why fecal samples were not collected from the group fed with diet supplemented with 5000mg/kg for microbiota analysis?
Suggestions 4. Tables: Lines 78 and 79 report that five additional diets were formulated by adding 250, 500, 750, 1000 or 5000 mg/kg C7. Whereas the tables in the results section do not show the effects of all the formulated diets. Some of the tables show the effects of diet supplemented with 250, 500, 750 and 1000 mg/kg supplementation of the additive but not the 5000 mg/kg (table 4, 6, 7 and 8). on the other hand, tables 5, 9, 10 shows the effects of diet supplemented with only two levels of the additive (500 and 5000 mg/kg) but not the others. Why the effects of all the experimental diets have not been reported in the results?
Reply: Thank you for your nice advice. The questions above could be replied together. The 5000mg/kg dose was set mainly to explore the tolerance of weaned piglets to C7. Therefore, the safety parameters of high dose C7 were mainly tested. The other groups were set up to assess the effectiveness of C7 and therefore varied in the measures examined.
Reference:
General Office of the Ministry of Agriculture of the People's Republic of China:
“Guidelines for Effectiveness Evaluation of Feed and Feed Additives for livestock and Poultry Target Animals (Trial)”
“Guidelines for Tolerance Evaluation of Feed and Feed Additives for Livestock and Poultry Target Animals (Trial)”
Suggestions 5. Lines 226, 228, 235 and 243: Please check numbering of the tables. Two tables have been numbered as table 8 and two tables have been numbered as table 9.
Reply: We feel sorry for the vague statement. The table is long and therefore split into two pages when uploaded using the word version, the following table is a continuation of the above. We will pay attention to this issue in the following layout.
Suggestions 6. Lines 325-332: These lines are suitable for introduction section.
Suggestions 8. Discussion: This section is week. Please improve the section.
Suggestions 9. Lines 390-391 & 395-396. This information is repeating in simple summary and abstract. Please avoid repetition.
Suggestions 10. Conclusions: Conclusions should be improved and re-written in connection with the specific findings of the study.
Reply: Thank you for your rigorous comment. Follow your suggestion, we revised introduction and discussion parts, we re-written the introduction and discussion, please check in the revised manuscript. We gratefully appreciate for your valuable suggestion.
Suggestions 7. Lines 347-352: Please provide a context of these lines with reference to this study.
Reply: We cite literature to show that immune immaturity at weaning and weaning stress can lead to a decline in the health status of piglets. Our findings showed that C7 supplementation increased serum IgG and anti-inflammatory cytokine IL-10 levels, decreased proinflammatory cytokine TNF-α levels. Supplementation with C7 improved some immune indicators, which may help alleviate the stress-induced negative state. As mentioned earlier, we have rewritten the discussion section and added relevant content, please check it out in the revised manuscript.
Reviewer 3 Report
The findings presented in the paper are interesting and will contribute to improving the welfare of piglets on commercial farms. The paper requires some minor modifications:
1. Authors state that approval from the Institutional review board was obtained, the approval number should be provided.
2. More information on why specific doses of Microcin C7 were chosen should be included.
3. Authors state 'The effect of C7 in 390 alleviating diarrhea may be related to its selective regulation on specific microbial taxa' - it is unclear what the authors mean by 'selective regulation' and should be explained further.
4. It would be good to include a statement about future studies in the conclusion.
Author Response
We would like to thank the reviewers for your professional comments and suggestions, which are very helpful in improving our manuscript. Based on the instructions provided in your letter, we uploaded the file of the revised manuscript. Appended to this letter is our point-by-point response to the comments raised by the reviewers.
Suggestions 1. Authors state that approval from the Institutional review board was obtained, the approval number should be provided.
Reply: Thank you for your rigorous comment. We have added the corresponding approval number, please check in the revised manuscript.
Suggestions 2. More information on why specific doses of Microcin C7 were chosen should be included
Reply: Thank you for your nice advice. C7 used in the experiment is provided from the AGELESS bio-tech co.,ltd. The company provides a reference range for the appropriate dose. According to the guidelines of the Ministry of Agriculture, we determined the doses.
Reference:
General Office of the Ministry of Agriculture of the People's Republic of China:
“Guidelines for Effectiveness Evaluation of Feed and Feed Additives for livestock and Poultry Target Animals (Trial)”
“Guidelines for Tolerance Evaluation of Feed and Feed Additives for Livestock and Poultry Target Animals (Trial)”
Suggestions 3. Authors state 'The effect of C7 in 390 alleviating diarrhea may be related to its selective regulation on specific microbial taxa' - it is unclear what the authors mean by 'selective regulation' and should be explained further.
Reply: We gratefully appreciate for your nice comment. We feel sorry for the vague statement.
Firstly, the microbiota structure of the C7 supplementation groups were negatively correlated with diarrhea incidence on d 14. Secondly, applying Spearman correlation analysis between the most differential microorganisms and diarrhea incidence, we found norank_f_Selenomonadaceae and Christensenellaceae were significantly and negatively associated with diarrhea incidence, and their abundance significantly increased in the C7-treated groups. We therefore reasoned that C7 promotes gut health by selectively regulating specific microbial taxa.
Suggestions 4. It would be good to include a statement about future studies in the conclusion.
Reply: Thank you for your nice advice. We are very sorry for our negligence of the statement about future studies. we re-written the discussion part, please check in the revised manuscript. We gratefully appreciate for your valuable suggestion.
Round 2
Reviewer 2 Report
Although the authors have attempted to provide responses to the comments already given, however, some of my earlier comments still needs to be addressed and manuscript should be improved accordingly. The comments are:
Please clarify why fecal samples were not collected from the group fed with diet supplemented with 5000mg/kg for microbiota analysis?
Tables report that five additional diets were formulated by adding 250, 500, 750, 1000 or 5000 mg/kg C7. Whereas the tables in the results section do not show the effects of all the formulated diets. Some of the tables show the effects of diet supplemented with 250, 500, 750 and 1000 mg/kg supplementation of the additive but not the 5000 mg/kg (table 4, 6, 7 and 8). on the other hand, tables 5, 9, 10 shows the effects of diet supplemented with only two levels of the additive (500 and 5000 mg/kg) but not the others. Why the effects of all the experimental diets have not been reported in the results? In response to these comments, the authors replied that:
Reply: Thank you for your nice advice. The questions above could be replied together. The 5000mg/kg dose was set mainly to explore the tolerance of weaned piglets to C7. Therefore, the safety parameters of high dose C7 were mainly tested. The other groups were set up to assess the effectiveness of C7 and therefore varied in the measures examined.
Reference:
General Office of the Ministry of Agriculture of the People's Republic of China:
“Guidelines for Effectiveness Evaluation of Feed and Feed Additives for livestock and Poultry Target Animals (Trial)”
“Guidelines for Tolerance Evaluation of Feed and Feed Additives for Livestock and Poultry Target Animals (Trial)”
Whether the referred guidelines bars reporting of safety evaluation data?
Suggestions 5. Lines 240, 242, 250 and 258: Please check numbering of the tables. Two tables have been numbered as table 8 and two tables have been numbered as table 9.
Suggestions 6. Lines 325-332: These lines are suitable for introduction section.
Suggestions 8. Discussion: This section is week. Please improve the section.
Suggestions 9. Lines 390-391 & 395-396. This information is repeating in simple summary and abstract. Please avoid repetition.
Suggestions 10. Conclusions: Conclusions should be improved and re-written in connection with the specific findings of the study.
Authors Reply: Thank you for your rigorous comment. Follow your suggestion, we revised introduction and discussion parts, we re-written the introduction and discussion, please check in the revised manuscript. We gratefully appreciate for your valuable suggestion.
The earlier comments 6, 8, 9 and 10 have not been addressed properly. Please address these comments properly and improve the manuscript accordingly.
The authors have made some revisions in the manuscript. However, there are irrelevant statements in the introduction and discussion sections that should be replaced with the literature relevant to the present study. The authors should focus on discussing the results of the particular study in the discussion section instead of giving general statements. Conclusions should also be based on the particular findings of the research conducted.
Author Response
Suggestions 1. Whether the referred guidelines bars reporting of safety evaluation data?
Reply: There is no provision in the guidelines to prohibit publication of the data. In general, data should be published after passing expert review, which we did.
Suggestions 2. The earlier comments 6, 8, 9 and 10 have not been addressed properly. Please address these comments properly and improve the manuscript accordingly.
Reply: Thank you so much for your careful check. According to your comments, we have moved the sentences in discussion to the introduction (line 50-53) and revised the repeated sentences in the conclusion (line 441-442; line 447-449). Thank you for your valuable modification suggestions on our article, which is very helpful for the improvement of our article level.
We apologize for the poor language of our manuscript. We worked on the manuscript for a long time and the repeated addition and removal of sentences and sections obviously led to poor readability. We thank the reviewer of pointing out this issue and sincerely hope that our new version will improve it.
We tried our best to improve the manuscript and made some changes in the manuscript. These changes will not influence the content and framework of the paper. We appreciate for reviewers’ warm work earnestly, and hope that the correction will meet with approval. Once again, thank you very much for your comments and suggestions.
Round 3
Reviewer 2 Report
.